DOI: 10.1038/s41467-017-02509-w · **OPEN**

# Transient rotation of photospheric vector magnetic fields associated with a solar flare

Yan Xu[1,2,3], Wenda Cao[2,3], Kwangsu Ahn[2], Ju Jing[1,2,3], Chang Liu [1,2,3], Jongchul Chae[4], Nengyi Huang[1,3], Na Deng[1,2,3], Dale E. Gary[3] & Haimin Wang[1,2,3]

As one of the most violent eruptions on the Sun, flares are believed to be powered by magnetic reconnection. The fundamental physics involving the release, transfer, and deposition of energy have been studied extensively. Taking advantage of the unprecedented resolution provided by the 1.6 m Goode Solar Telescope, here, we show a sudden rotation of vector magnetic fields, about 12–20° counterclockwise, associated with a flare. Unlike the permanent changes reported previously, the azimuth-angle change is transient and cospatial/temporal with H$\alpha$ emission. The measured azimuth angle becomes closer to that in potential fields suggesting untwist of flare loops. The magnetograms were obtained in the near infrared at 1.56 μm, which is minimally affected by flare emission and no intensity profile change was detected. We believe that these transient changes are real and discuss the possible explanations in which the high-energy electron beams or Alfve'n waves play a crucial role.

[1] Space Weather Research Lab, New Jersey Institute of Technology, 323 Martin Luther King Blvd, Newark, NJ 07102-1982, USA. [2] Big Bear Solar Observatory, New Jersey Institute of Technology, 40386 North Shore Lane, Big Bear City, CA 92314-9672, USA. [3] Center for Solar-Terrestrial Research, New Jersey Institute of Technology, 323 Martin Luther King Blvd, Newark, NJ 07102-1982, USA. [4] Astronomy Program, Department of Physics and Astronomy, Seoul National University, Seoul 151-747, Republic of Korea. Correspondence and requests for materials should be addressed to Y.X. (email: yan.xu@njit.edu)

It is well accepted that many solar flares, and other violent eruptions such as coronal mass ejections (CMEs), result from magnetic reconnection occurring in the corona. However, many manifestations are visible in the lower solar atmospheres, i.e., the chromosphere and photosphere. In addition to the radiation, significant changes of the magnetic fields have been observed in the literature[1–6]. Most of them are permanent changes, while transient changes are rarely reported, and are suspected of being instrumental artifacts.

Permanent magnetic changes are irreversible phenomena of photospheric fields in reaction to the flare impacts, usually during strong flares greater than M-class. Shear flows measure horizontal motions of magnetic features along both sides of the magnetic polarity inversion line (PIL). Previous observations show the spatial correlation between strong shear flow and flare emission[2,7]. More importantly, shear flow can drop significantly after

the flare[3,5,8], indicating that a certain amount of magnetic free energy has been released. Tilt angle, also known as inclination angle, is determined by the ratio between vertical and horizontal magnetic components. During flares, the reconnection rearranges the topology of magnetic loops and the tilt changes consequently. Direct measurements of vector fields have shown that horizontal fields increase near the PIL and decrease in the nearby penumbral regions during flares[6,8,9]. As a consequence, sudden intensity changes of magnetic features (usually the penumbra) are observed[1,10–12]. Bodily rotation is one of the intrinsic properties of sunspots or sunspot groups first observed by Evershed (1910)[13], which are usually gradual and continue during the entire lifetime. On the other hand, rapid rotations associated with X-class flares were reported[5,14] and attributed to the torque introduced by the change of horizontal Lorentz force[15]. Using the data with higher spatiotemporal resolution obtained by the 1.6 m Goode Solar

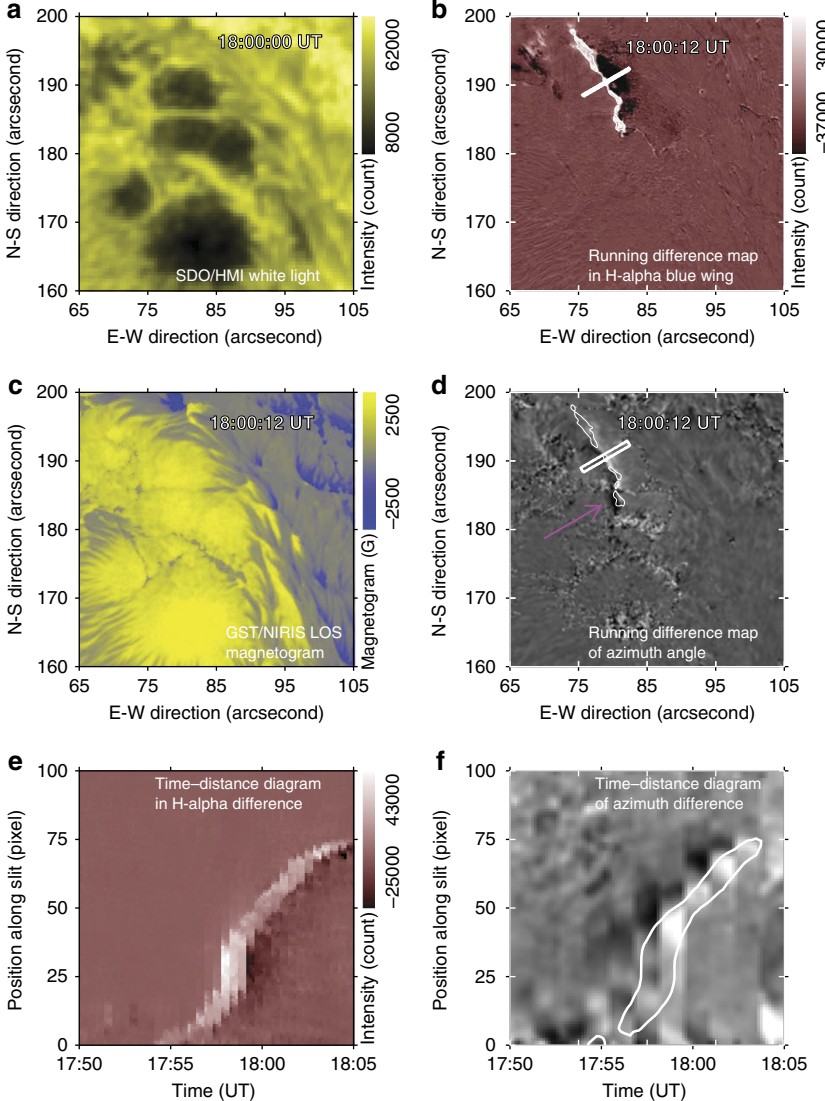

**Fig. 1** Azimuth angle changes in association with Flare emission. All of the four images (first and second rows) were taken simultaneously at the flare peak time (18:00 UT) in a common FOV of 40″ by 40″. **a** SDO/HMI white light map. Running difference image in Hα blue wing (−1.0 Å), showing the eastern flare ribbon in (**b**). The bright part is the leading front and the dark component is the following component. **c** The GST/NIRIS LOS magnetogram, scaled in a range of −2500 (blue) to 2500 G (yellow). Running difference map of azimuth angle generated by subtracting the map taken at 17:58:45 UT from the one taken at 18:00:12 UT in (**d**). The dark signal pointed to by the pink arrow represents the sudden, transient increase of azimuth angle at 18:00:12 UT. The white contours outline 60% of the maximum emission of the Hα ribbon front. **e** Time–distance diagram of Hα difference maps. The slit position is shown in (**b**). The time period is from 17:50 UT to 18:05 UT. **f** Time–distance diagram of azimuth difference maps. The slit position is shown in (**d**). The time period is from 17:50 UT to 18:05 UT. The white contours outline 15% of the maximum emission of the Hα ribbon front in (**e**)

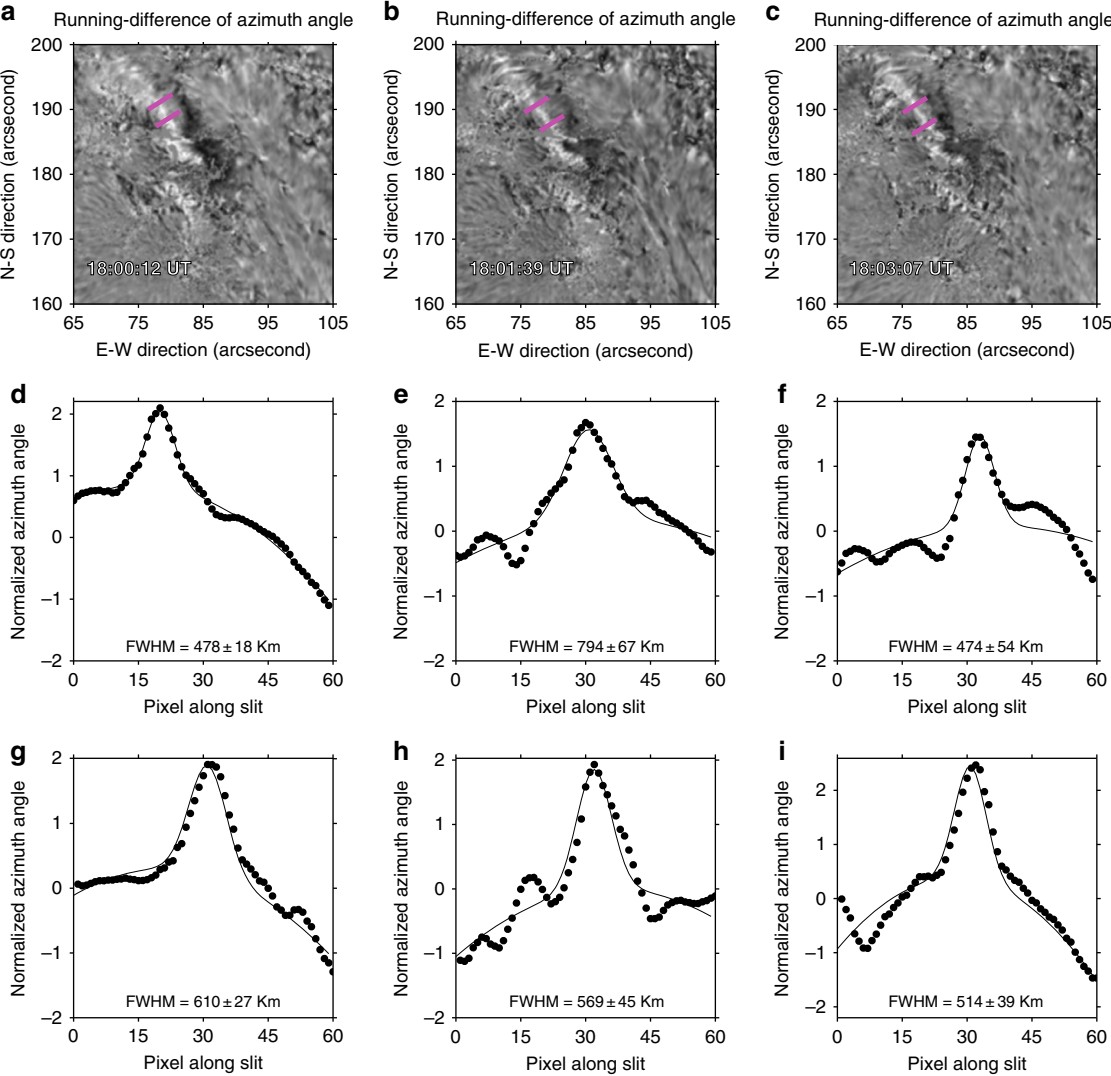

**Fig. 2** Characteristic sizes of the region of azimuth angle deviation. **a**–**c** Sparse running difference maps of azimuth angles, taken at three representative times. **d**–**f** Azimuth angle profiles along the top slit shown in each image in (**a**–**c**) and the corresponding Gaussian fits. **g**–**i** Azimuth angle profiles along the lower slit shown in each image (**a**–**c**) and the corresponding Gaussian fits. The FWHM, derived from the fitting, is used as the ribbon width of azimuth change, which is about 570 km on average

Telescope (GST, formerly known as the New Solar Telescope[16]) at Big Bear Solar Observatory (BBSO[17]), the sudden rotation is found to be nonuniform and synchronous with the flare ribbon propagation[18].

In contrast to the stepwise temporal profile in permanent changes, the characteristic profile of a transient change is more like a δ function in time. Transient changes were rarely observed in the literature. An example is the magnetic anomaly or magnetic transient, a temporal reversal of magnetic polarities measured simultaneously with flare emission, first reported by[19,20] using BBSO data. The plausible explanation is that the profile of the Fe I line at 5324 Å, used for the magnetic measurements, turned from absorption to emission due to the flare heating at lower layers of solar atmosphere[19]. From space-based observations, magnetic anomalies were reported during an X 5.6 flare observed by the Michelson Doppler Imager on-board Solar and Heliospheric Observatory (MDI/SOHO)[21] and an X 2.2 flare observed by the Helioseismic and Magnetic Imager on-board Solar Dynamics Observatory (HMI/SDO)[22]. The authors drew similar conclusions that the polarity reversal is a consequence of the line profile change. Thus, the magnetic anomaly/transient is

not intrinsic to the Sun, but an artifact in magnetic measurements due to the change of line profile.

In this paper, we present 1.56 μm vector magnetograms with the highest cadence and resolution ever obtained, which are much less subject to line profile changes, yet reveal a sudden increase of the azimuth angle. An M6.5 flare was well observed on June 22, 2015 with BBSO/GST using multiple channels, including the Visible Imaging Spectrometer (VIS) tuned to the Hα line, a broadband filter imager (BFI) tuned to a continuum near the TiO line at 7057 Å, and the Near InfraRed Imaging Spectropolarimeter (NIRIS) providing vector magnetograms using the Fe I line at 1.56 μm. The image scale of vector magnetograms is about 0″.083 pixel$^{-1}$ and the cadence is about 90 s for a full set of Stokes measurements. The Fe I Stokes profiles are measured at 40 different spectral positions, which is much higher than that on space-based spectropolarimeters, such as MDI and HMI, and permits a check of possible changes in line profile due to enhanced emission, which are not seen.

The flare started around 17:39 UT and peaked at 18:23 UT in GOES 1−8 Å soft X-rays. It lasted for several hours and our interest in this study focuses on the initial phase within the core

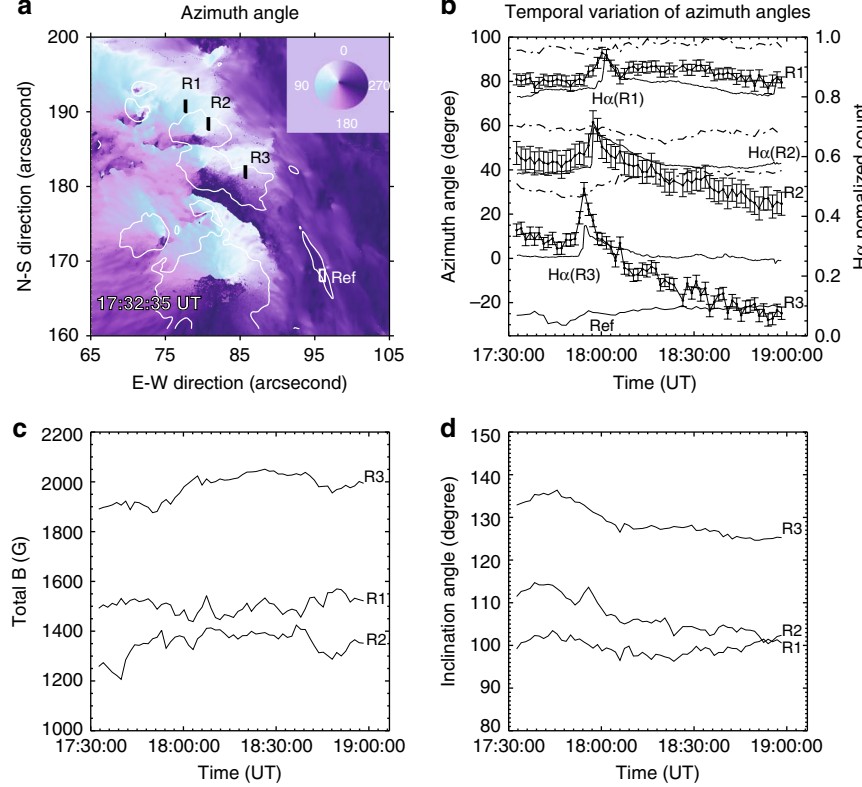

**Fig. 3** Temporal evolution of azimuth angle deviation. **a** Azimuth angle map taken before the flare at 17:32:35 UT. Three slits are put on the regions of interest (R1-3), plus a reference region in the lower right corner. The white contours outline the sunspot umbral areas (>1800 G). **b** The curves with error bars are the temporal variation of averaged azimuth angle within regions of R1-3. The uncertainties are estimated using the standard deviation of the preflare data points. The peaks are more than three times of the uncertainties rendering themselves statistically significant. The flare time is determined by the Hα light curve, for instance, the dashed line is the Hα light curve of R3, in which the peak matches with azimuth angle peak in R3. All Hα light curves are in natural log space and self-normalized to their peak emission. In the bottom, the temporal variation of the azimuth angle in the reference region is plotted, which is manually increased by 50° to match the plotting range (50–190°). The dotted-dash curves are the azimuth angles of extrapolated potential fields that remain certain levels above the azimuth angles of real fields. **c** Temporal variation of averaged magnetic flux strength within the representative areas. **d** Temporal variation of averaged inclination within the representative areas

region of the flaring areas. The host active region NOAA 12371 was close to the solar disk center at that time and therefore the geometric projection effect is small and was corrected easily. From the time sequence of azimuth maps, we clearly see a ribbon-like structure moving cospatially and cotemporally with the flare emission in the Hα line. On average, the azimuth angles increased by about 12–20°, indicating that the local magnetic fields rotated counterclockwise. In contrast to the permanent changes of magnetic field, the azimuth change is a transient variation, which restored quickly to its original value after the flare ribbons swept through. By reviewing the existing models, such as Alfve'n waves and induced magnetic fields, we found that they may play important roles but can not solely explain the observation.

## Results

**Overview of the flare.** Two major flare ribbons are identified in the core area of the flare. The eastward-moving ribbon resides in the area of positive magnetic polarity and the westward-moving ribbon propagates in the region with negative magnetic polarity. We focus on the eastern ribbon, where azimuth angle increases are much more obvious. A change of azimuth angle can also be identified associated with the conjugate flare ribbon within negative magnetic fields, but it is dispersed and too weak to be precisely measured. The elongated flare ribbons represent multiple footpoints of parallel flare loops. For each individual loop, the change of azimuth angle on its footpoints indicates a twisting

or untwisting of the loop. To determine whether the twist of the loop is increased or decreased, the difference between the azimuth angles of the measured magnetic field and that of the extrapolated potential fields is calculated, in which the latter was extrapolated with the Fourier transformation method[23,24]. As we see below, the vertical component of magnetic field ($B_z$) and the extrapolated potential field ($B_p$, derived from $B_z$), remains nearly constant during the flare. Therefore, the comparison of transverse components of observed and extrapolated fields, represented by the azimuth angle, can in principle indicate the variation of the twist.

**Characteristics of the azimuth angle variation.** Supplementary Movie S1 shows the time sequence of azimuth angle within a field of view (FOV) of the flare core region, running from 17:32:35 UT to 18:58:19 UT. It is clear that a ribbon-like feature propagates from the right to the left. This disturbance of azimuth angle along a narrow ribbon is cospatial and cotemporal with the flare emission seen in Hα. In Fig. 1d, a running difference map of azimuth angle is shown at 18:00:12 UT. The dark feature indicated by the pink arrow represents the change of azimuth angle and the white contours show the leading front of the Hα emission, indicating a close relationship with precipitating electron beams[25]. The slight offset of about 300~500 km could be a projection effect, because the formation height of Hα is about few thousand km higher than the formation height of the NIR line at

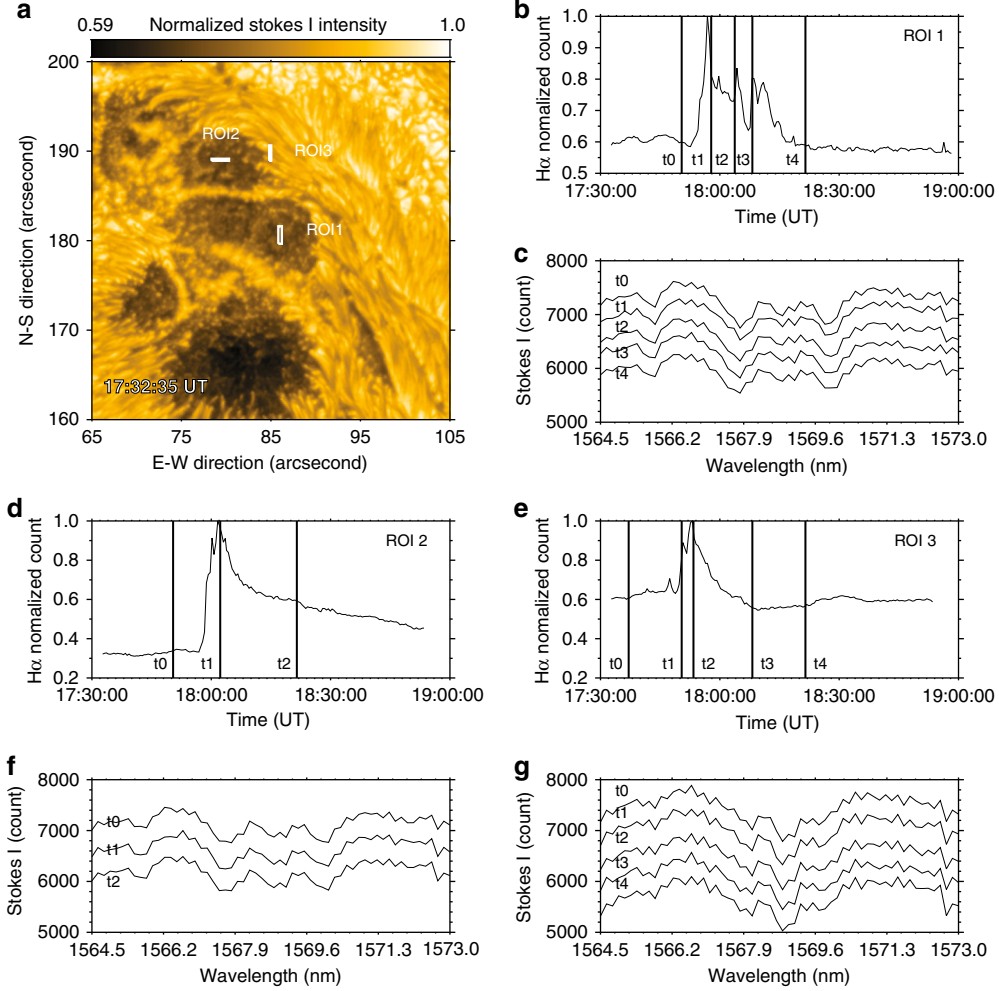

**Fig. 4** Intensity profiles of the NIR line at 1.56m during the flare **a** Stokes I component taken at 17:32:35 UT. The intensity is normalized to the maximum count, as shown in the color bar. Three representative areas are marked using white boxes (ROI1, ROI2, and ROI3). **b** Hα light curve in ROI 1. The vertical lines indicates five time points before, during, and after the flare. The corresponding NIR intensity profiles (Stokes I) are plotted in (**c**), from which we see almost identical line profiles indicating that the flare heating almost has no effect in this deep layer of solar atmosphere. **d** Hα light curve in ROI 2. The corresponding NIR intensity profiles at t0, t1, and t2, are plotted in (**f**). **e** Hα light curve in ROI 3. The corresponding NIR intensity profiles at t0-t4 are plotted in (**g**)

1.56 μm. Other panels show the SDO/HMI white-light image in Fig. 1a, a running difference image of Hα blue wing (−1.0 Å) in Fig. 1b, and LOS magnetogram derived from NIRIS data in Fig. 1c. All the images are within the same FOV, where the ribbon of interest resides. To examine the azimuth change, we resolved the 180° azimuthal ambiguity in the transverse field using the minimum-energy method[26], and removed the projection effects by transforming the vector magnetogram from observational image plane to heliographic-cartesian coordinate. Figure 1e presents a time–distance diagram of the running difference Hα images. The slit is 3-pixel wide and its position can be found in Fig. 1b. The bright feature represents the ribbon front of Hα emission, similar to the one in Fig. 1b. In Fig. 1f, we display the time–distance diagram of running difference images of the azimuth angle. The slit width is 9 pixels, because the cadence of the vector magnetogram is 90 s, which is about 3 times of the cadence of Hα images. The white contour indicates the location of Hα ribbons in Fig. 1e. It shows a good correlation between Hα emission and azimuth-angle change at different times. Such a correlation does not vary much at different slit positions.

The characteristic sizes of ribbon-like features are fundamental parameters. For instance, the ribbon front, which is the precipitation site of electrons, is found to be very narrow[27,28].

We follow the method described in Xu et al.[25] and Jing et al.[27] to measure the width of the azimuth ribbon, as shown in Fig. 2. We use sparse running difference maps (the reference image is taken several frames prior to the target) to minimize the background noise. On average, the width of the region of azimuth angle deviation is about 570 km, which is comparable to the size of the dark ribbon (340–510 km) in helium at 10830 Å[25]. It is not easy to measure the ribbon length quantitatively, as the ribbon is segmented and noisy near the two ends. We estimate the length manually using the image taken at 18:00:12 UT and the result is about 13,300 km.

**Temporal evolution of the azimuth angle change and correlation with Hα emission.** In order to study the temporal evolution of the disturbance, three representative regions (R1, R2, and R3, marked using white color) are selected on the propagating path of the azimuth transient, as shown in Fig. 3a. These representative slits are selected in the middle of the ribbon and away from the sunspot boundary, where the magnetic fields are also affected by the sudden sunspot rotation[18] in a more gradual manner. The corresponding temporal profiles are plotted in Fig. 3b. The uncertainties are estimated using the points prior to the initiation of the flare. For instance, the standard deviation of the first 12

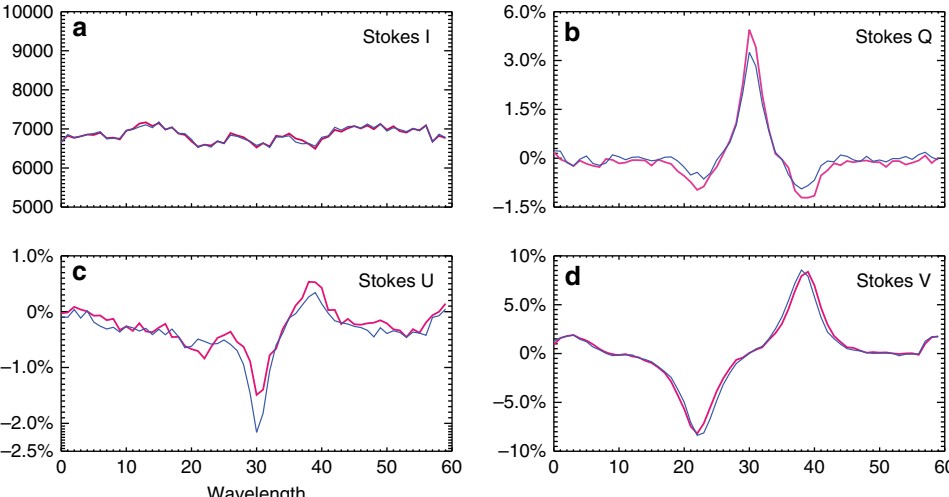

**Fig. 5** Stokes profiles before and during the flare. Stokes components (I, Q, U, and V) taken near R3 before (blue) and during (pink) the flare. **a** Stokes I. **b** Stokes Q. **c** Stokes U. **d** Stokes V. It is clear that the Stokes I and V components remain almost unchanged but Q and U components are significantly affected during the flare

points is used as the error in R1. Let us use R3 as an example. The average azimuth angle suddenly increases by about 20°, from the preflare value of 11.7° to the flare peak time value of 31.5°, with an uncertainty of 6.6°. Therefore, the azimuth peak is significant as it is about three times of the uncertainty. In particular, we see that the azimuth peak coincides with the Hα emission (dashed curve), based on the results shown in Figs. 1 and 3. For the other two regions, R1 and R2, the horizontal field rotates by 12° and 18°, with uncertainties of 2.5° and 5.2°, respectively. Using the potential field extrapolation, the azimuth angle of potential field is determined and plotted as dot-dash curves in Fig. 3b. We see that the potential field azimuth remains at a certain level above the azimuth angle of the vector fields. Only at the flare peak time, the measured azimuth angle becomes closer to that in the potential field due to the sudden rotation. This is a 2D comparison, but is a good proxy of the 3D configuration, because the extrapolated potential fields are based on the measured vertical component, which did not vary like the azimuth angle did during the flare. Therefore, the difference of the magnetic shear between the measured field and the potential field can be represented by the azimuth angle, which is determined by the horizontal components of $B_x$ and $B_y$. In Fig. 3c, d, the total magnetic strength and the inclination angle are plotted as a function of time, respectively. These curves contain noise-induced fluctuations and no impulsive peak is identified as in the azimuth transient. An area (white box) is selected far away from the flare ribbons and is used as a reference in comparison to the regions with significant azimuth angle changes. We see irregular fluctuations but certainly no obvious peak associated with flare emission.

## Discussion

In summary, we present a clear transient change of azimuth angle, associated with propagating flare ribbons which are footpoints of 3D magnetic loops. The major results are as follows: 1) The local magnetic vectors rotated about 12–20° simultaneously with flare emission. 2) The strong correlation between the azimuth transient and flare ribbon front indicates that the energetic electron beams are very likely to be the cause. 3) The measured azimuth angle becomes closer to that in the potential field, indicating a process of energy release (untwist) of the flare loop.

First, the observed field change is different from the magnetic anomaly reported previously. In that scenario, the profile of the spectral line used to measure the magnetic fields has changed. In

our case, the line profiles remain in absorption and unchanged during the flare, as shown in Fig. 4. It is not surprising as the spectral line used by NIRIS is the Fe I line at 1.56 μm, which is formed very deep in the photosphere where almost no flare heating can reach except in some extremely strong flares[29]. In addition, we investigate the polarized raw data before inversion is done. On the Stokes I, Q, U, and V maps, as shown in Fig. 5, an area similar to R3 is selected and the corresponding profiles before and during the flare are plotted. As we can see, the Stokes I and V components, in which V represents the circular polarization and determines the LOS magnetic fields, remain almost identical. However, the Stokes Q and U components that determine the transverse fields, vary significantly. Therefore, we believe that azimuth change is real from Stokes Q and U components and not affected by either flare emission or circular polarizations.

This is the first time that transient field rotation is observed. We attempt to explain the effect by considering several existing models, which are discussed below.

Magnetic field induced by the electron beams is the most straightforward and intuitive model. The basic idea is that the penetrating electrons produce induced magnetic fields, which act on the original fields such that the combined fields point to new directions. This is equivalent to a field rotation. The downward precipitation of electrons, with negative charges, is equivalent to an upward current. According to Ampère's circuit law, self-induced magnetic field is generated around the ribbon. The ribbon width is much smaller than its length, so the latter can be treated as a half-infinite wall. Therefore, to the left side of the ribbon front, the self-induced magnetic field points to the south (in solar coordinates) and the overall field will rotate counter-clockwise. To the right side, the overall field will rotate oppositely, which however will be balanced by the self-magnetic field of the trailing electron beams that have decreased but have not been turned off. We can estimate the required current $I$ using Ampère's law, $\oint \mathbf{B} \bullet \mathbf{ds} = \mu_0 I$, in which the integration loop is $2l$ (two times the ribbon length, which is about $2 \times 10^7$ m). In order to induce a magnetic field of order 100 G, the required current is about $1.6 \times 10^{11}$ A, or an electron flux of $10^{30}$ s$^{-1}$, a tiny fraction of the total electron flux (~$10^{35}$) with energy greater than 20 keV derived from the RHESSI HXR spectra. Whether the azimuth angles of the combined fields increase or decrease is determined by the relative orientation between the original fields and the flare ribbon. If this angle is larger than 90°, an increase is seen in front of

the flare ribbon. The rotation effect is canceled out behind the flare ribbon by the following opposite rotation effect. Since the fields point outward from a sunspot center, when the ribbon passes through the sunspot, the relative orientation changes and the azimuth angle should decrease. However, such an increase–decrease pattern is not observed. Although the magnitude of induced field matches with observations, the direction does not match.

The second model considers the effect by downward-drafting plasma[30], which in our case is the precipitating electron beam. The authors modeled a scenario in which the cooling plasma moving down from the top of hot granules amplifies the magnetic twist when entering into denser layers, which is similar to our case. However, we see that the flare loops become less twisted. Nevertheless, this model suggests that the hydrodynamic effects may be coupled with electrodynamic effects to affect the preflare magnetic fields.

By analyzing and modeling the H$\alpha$ and H$\beta$ data, Hénoux and Karlický[31] found that emission lines can be polarized linearly by multiple effects, such as electron beams, return current, and filamentary chromospheric evaporation. They found that the degree of linear polarizations was about 3~9%. However, in our case, there was no emission detected in the NIR line at 1.56 μm. The NIR line intensity profile remains in absorption during the flare. In addition, the azimuth change, or say the enhanced linear polarization was only found in front of the propagating flare ribbons in our event. But the polarized signal was identified on both sides of the flare ribbons in Hénoux and Karlický[31]. Therefore, again, we cannot draw a conclusion based on their model.

Alfvén waves[32] also have impacts on the magnetic fields. They are well known in heating the corona[33], accelerating electrons during flares[34], and solar winds[35]. Alfvén waves can be generated by magnetic reconnection during flares[34,36]. In open magnetic field regions, for instance, in solar wind, Alfvén waves were found in untwisted field lines both in observations[37] and simulations[38]. For closed field regions, such as the flare loops, Fletcher et al.[34] presented the large-scale Alfvén wave pulses, which accelerate electrons locally. Within nonuniform plasma, Alfvén waves appear as torsional waves[39], which can create rotational perturbations of the plasma and the magnetic fields frozen in the plasma[40]. The perturbations are usually torsional oscillations[41], which should appear periodically but were not observed in our event. However, in the deep atmosphere, these waves can be damped locally by ion-neutral or resistive damping[42] and therefore, only the effect of the initial pulse is observed as a transient event. Alfvén waves are plausible candidates, but most previous modeling was done for coronal flux tubes and no quantitative description is available for their effects on photospheric magnetic fields.

In conclusion, the observed field change cannot be explained by existing models. The new, transient magnetic signature in the photosphere that we describe in this paper offers a new diagnostic for future modeling of magnetic reconnection and the resulting energy release. Such observations require high cadence and high resolution. Our results motivate further observations using GST and the Daniel K. Inouye Solar Telescope (DKIST) in probing the mystery of solar flares.

## Methods

**Magnetic inversion**. After dark and flat field correction, the cross talk is removed by measuring the effect of optical elements from the telescope to the detector. Pure states of polarization are fed into the light path and their response at the detector tells how the incoming polarization from the Sun would be changed by the optics[43] (and references therein). After careful elimination of the cross talk among Stokes Q, U, and V, the NIRIS data undergo Milne-Eddington inversion to fit the Stokes line profiles based on an atmospheric model. The source function with respect to optical depth is simplified to a first-order polynomial. From results, several key

physical parameters can be extracted—$B_{tot}$, azimuth angle, inclination, Doppler shift, and so forth. For successful fitting into ME-simulated profiles, initial parameters are precalculated to be in proximity to the observed Stokes profiles. This code was specifically designed for BBSO/NIRIS and written by Dr. J.C. using IDL language.

**180° ambiguity correction**. In order to streamline the analysis of vector magnetogram data, data-processing tools have been developed and implemented, including the 180° ambiguity resolution and NLFF coronal magnetic field extrapolation. The 180° azimuthal ambiguity in the transverse magnetograms is resolved using the minimum-energy algorithm that simultaneously minimizes both the electric current density $J$ and the field divergence $|\nabla \cdot \mathbf{B}|$[26]. Minimizing $|\nabla \cdot \mathbf{B}|$ gives a physically meaningful solution and minimizing $J$ provides a smoothness constraint. A magnetogram is first broken into small subareas with which to compute a force-free $\alpha$ parameter. Then, a linear force-free field is effectively constructed with which to infer the vertical gradients needed to minimize the divergence. Since the calculation of $J$ and $|\nabla \cdot \mathbf{B}|$ involves derivatives of the magnetic field, the computation is not local, the number of possible solutions is huge, and the solution space has many local minima. The simulated-annealing algorithm[44] is used to find the global minimum. This minimum-energy algorithm is the top-performing automated method among the present state-of-the-art algorithms used for resolving the 180° ambiguity[45].

**Data availability**. All the data used in the present study are available to the public. The BBSO/GST data, including the vector magnetograms and Hα data can be downloaded from http://bbso.njit.edu. The SDO/HMI white-light images and vector magnetograms can be downloaded from http://jsoc.stanford.edu. The extrapolation codes as used in this study can be obtained from http://www.lmsal.com/solarsoft.

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

## Acknowledgements

We thank the BBSO and SDO/HMI teams for providing the data. The BBSO operation is supported by NJIT, US NSF AGS 1250818 and NASA NNX13AG14G grants, and the GST operation is partly supported by the Korea Astronomy, Space Science Institute, Seoul National University, and by the strategic priority research programme of CAS with Grant Number XDB09000000. This work is supported by NSF under grants AGS 1250818, 1348513, 1408703, and 1539791, and by the NASA LWSTRT grants NNX13AF76G and NNX13AG13G, HGI grants NNX14AC12G and NNX16AF72G. W. C. was supported by NSF AGS-0847126. D.E.G. was supported by NSF AST-1615807. J. C. was supported by the National Research Foundation of Korea (NRF2012 R1A2A1A 03670387).

## Author contributions

Y.X. was the PI of this GST observation run, discovered the transient change of azimuth angle, performed data analysis and interpretation, and wrote/revised the manuscript. W. C. developed instruments of GST and coordinated the observation. K.A. performed the magnetic inversion. J.J. and C.L. helped with the correction of 180° ambiguity and deprojection. J.C. developed the magnetic inversion code and helped with the interpretation. N.H. and N.D. contributed to the image processing. D.E.G. contributed to the result interpretation and revising of the manuscript. H.W. provided overall scientific guidance, model discussion, and directed this research. All authors commented on the manuscript.

## Additional information

**Competing interests:** The authors declare no competing financial interests.

