## [Peer Review File · Nature Communications]

Reviewers' comments:

Reviewer #1 (Remarks to the Author):

The paper claims to have discovered a transient rotation of the vector magnetic fields during a solar flare using the unprecedented resolution observations by the 1.6-m New Solar Telescope at Big Bear Solar Observatory. While presented observations are novel and will be of interest to the community, the analysis of these observations is not very convincing.

1) While the authors claim that there is a correlation between the magnetic field azimuth and the ribbon locations, they do not present a clear side-by-side spatial and temporal comparison of the vector magnetic field azimuth and ribbon observations. Correlation analysis of the temporal profiles of ribbons and azimuth observations at the same location or an overlay of the ribbon-edge contours over the azimuth image at different times would be useful to evaluate their relationship, correspondent time delays, length of the discovered phenomenon. The video would also benefit from an overlay of the ribbon edges onto the azimuth and a correct time label (currently the time is wrong).

2) The whole argument regarding the magnetic field potentiality and the field becoming more untwisted is not very clear: no details are presented regarding how the potential magnetic field is evaluated and what are the error bars in the corresponding azimuthal components. If we assume that the field becomes more untwisted and the change in the azimuth is temporal, then it is not clear what makes the field to become more twisted right after the transient. Analysis of the active region non-potential versus potential energy could be helpful.

To summarize the paper would benefit from a more quantitative analysis of spatial and temporal changes in azimuth and ribbon observations to draw more convincing conclusions.

Reviewer #2 (Remarks to the Author):

The authors report on the apparently first observation of the transient change of the photospheric magnetic field in a large part of the flare neighborhood which, according to the authors, cannot be explained as being due to the observational artifacts. As the authors write, similar events have been observed in the past but it is their work which rules out the instrumental-artifact explanation.

The most significant observational evidence the authors present to support their hypothesis is the fact that the intensity profiles of the photospheric iron line around 1.56 μm does not show any significant change during the flare event. The profiles shown in Fig. 3 are indeed very similar to each other for multiple observations. In fact, the noise across the profiles seems to be practically identical for different observations taken minutes from each other which is a bit strange. Perhaps it is an instrumental effect but there is not explanation to it. The authors do not show profiles of other Stokes parameters from which the magnetic field azimuth has been determined and they do not mention which method of magnetic field inversion has been used. Since this information is critical for evaluation of the results, it is difficult to judge the claims of the manuscript.

The authors suggest a simple qualitative model of how the magnetic field induced by the energetic electron beams from the coronal reconnection site could be superposed with the magnetic field in the photosphere and how it could cause the observed changes in the magnetic field azimuth. Probably the main problem with this explanation is that the electron beams cannot even reach the photospheric layers where they are supposed to influence the photospheric magnetic field.

The observations presented in this manuscript may have a substantial scientific potential but since

a short letter like this cannot explain the essential details, I think the work should better be published in a detailed paper in a journal like *The Astrophysical Journal* or *Solar Physics*. In the present form, the paper does seem to provide convincing scientific evidence for its strong claims.

Re: Revisions for manuscript NCOMMS-17-08380T

Referee Report #1:

1) While the authors claim that there is a correlation between the magnetic field azimuth and the ribbon locations, they do not present a clear side-by-side spatial and temporal comparison of the vector magnetic field azimuth and ribbon observations. Correlation analysis of the temporal profiles of ribbons and azimuth observations at the same location or an overlay of the ribbon-edge contours over the azimuth image at different times would be useful to evaluate their relationship, correspondent time delays, length of the discovered phenomenon. The video would also benefit from an overlay of the ribbon edges onto the azimuth and a correct time label (currently the time is wrong).

Answer:

1) We modified Figure 1 to show the ribbon-edge contours in H-alpha over the azimuth angle image. The azimuth signal of variation is relatively weak, so the H-alpha contours can easily make the subtle azimuth variation disturbance invisible in the movie. Therefore, we do not use the H-alpha contours in the movie. But we checked the spatial correlation manually and display the H-alpha contours over azimuth angles in three different times below, from which we see the H-alpha contours are co-spatial with the dark features on the reversed difference maps of azimuth angle.

- 2) In addition, the temporal profile of H-alpha emission in R3 is over-plotted in Figure 2, upper right panel. The azimuth changes correlate with H-alpha emission very well for all the regions, including R1, R2 and R3. There is no detectable delay between the H-alpha emission and the change of azimuth angles. We choose not to plot the light curves of R1 and R2 to avoid an overly busy figure.
- 3) We add a figure (Fig. 2) and a paragraph to show the measurements of characteristic sizes of disturbance of the azimuth angle associated with ribbon front.
- 4) Sorry for the wrong time labels of the movie. It is now corrected.

2) The whole argument regarding the magnetic field potentiality and the field becoming more untwisted is not very clear: no details are presented regarding how the potential magnetic field is evaluated and what are the error bars in the corresponding azimuthal components. If we assume that the field becomes more untwisted and the change in the azimuth is temporal, then it is not clear what makes the field to become more twisted right after the transient. Analysis of the active region non-potential versus potential energy could be helpful.

Answer:

1) The potential extrapolation is performed with the code 'fff.pro' (by T. Metcalf) which should be found in SSW package ([http://www.heliodocs.com/php/xdoc_print.php?file=\\$SSW/packages/nlfff/idl/fff.pro](http://www.heliodocs.com/php/xdoc_print.php?file=$SSW/packages/nlfff/idl/fff.pro)). According to the modification history of the code, this code is a modification of Sandy and Yuhong's 'potential92.pro'.

2) For each temporal profile of azimuth, the error bar is determined by its fluctuations before the flare time. For instance, we used the standard deviation of the 1st 12 points of the temporal profile in R1 as the error.

3) We assume the loops with a given twist are stable before the flare. During the flare, due to some perturbation (like an oscillation in the magnetic field or other disturbance) the loops became temporarily less twisted but unstable. Therefore, after the perturbation the loops will return back to the pre-flare state. We think the free energy analysis, comparison of non-potential vs. potential, would be suitable for large scale structures, but the flare loops are in much smaller scales.

Referee Report #2:

The most significant observational evidence the authors present to support their hypothesis is the fact that the intensity profiles of the photospheric iron line around 1.56 um does not show any significant change during the flare event. The profiles shown in Fig. 3 are indeed very similar to each other for multiple observations. In fact, the noise across the profiles seems to be practically identical for different observations taken minutes from each other which is a bit strange. Perhaps it is an instrumental effect but there is no explanation to it. The authors do not show profiles of other Stokes parameters from which the magnetic field azimuth has been determined and they do not mention which method of magnetic field inversion has been used. Since this information is critical for evaluation of the results, it is difficult to judge the claims of the manuscript.

Answer:

1) The Stokes I components are similar but with small differences. It is because they are actually averaged with a certain areas so that the individual fluctuations are smoothed. In addition, due to the strong magnetic fields, this line has large Zeeman

splitting and becomes very shallow.

2) In order to make the claim more convincing, we re-investigate the 'raw' data before Stokes inversion. We add a new figure (Fig. 5) to show the Stokes components before the flare and during the flare. From these new plots, we see almost identical Stokes I and V components before/after the flare, but clear different profiles of Stokes Q and U which determine the transverse fields.

3) Milne-Eddington method is used for the inversion. Now it is clarified in the manuscript. Actually we also compared NIRIS magnetograms with the vector magnetograms taken by SDO/HMI and Hinode. The correlation is pretty high as seen in their figure (above 90%, Wang et al., 2017).

The authors suggest a simple qualitative model of how the magnetic field induced by the energetic electron beams from the coronal reconnection site could be superposed with the magnetic field in the photosphere and how it could cause the observed changes in the magnetic field azimuth. Probably the main problem with this explanation is that the electron beams cannot even reach the photospheric layers where they are supposed to influence the photospheric magnetic field.

Answer: We agree that the electron penetration depth may be a question. But for producing the induced field, the electrons are not required to penetrate to the

photosphere although induced field decreases inversely proportional to the distance to source current. On the other hand, we found that this simple explanation is instructive but doesn't explain all the observational facts. For instance, whether the azimuth angle increases or decreases depends on the relative orientation between the original fields and the direction of ribbon motion. When the ribbon is on the right-hand side of the sunspot, we should see azimuth increases. But when the ribbon moves to the left-hand side of the sunspot, we should see a decrease, which is not seen in the observations. Therefore, we choose to discuss several potential models instead of a very preliminary one. As this is the very first observation of such kind of phenomena, we'd rather leave the question open for future observations and modeling. The changes in discussion are highlighted.

Other changes:

1. The New Solar Telescope (NST) has been renamed to Goode Solar Telescope (GST) in this July.
2. The absolute values of all of the azimuth angles have been changed (decreased) by 90 degrees. This is due to the reference axis used was not consistent with that used by SDO/HMI or Hinode. Now we made ours consistent with the space data. The results of azimuth angle changes are not affected by this.

Reviewers' comments:

Reviewer #1 (Remarks to the Author):

General comments:

1. The correspondence between flare ribbons and the azimuth disturbances has been demonstrated at only one time (Figure 1). Please either modify the manuscript to reflect the fact that you found correspondence at only one time or add similar to Figure 1 comparisons at other times and locations that would demonstrate that ribbons and azimuth disturbances have the same dynamics. For example, a side-by-side comparison of ribbon vs azimuth space-time plots across the slit perpendicular to the ribbon front or comparison of 1) color map of ribbon propagation with color demonstrating the time of the initial ribbon brightening overlaid on the contours of the magnetic field and 2) color map of azimuth disturbance with color demonstrating the time of the initial azimuth disturbance with contours of the magnetic field, would be helpful.

2. Assumption that the field becomes more potential implies that the angle between the measured field and the potential field decreases. However in Figure 3 you only compare the azimuths of the two, not the azimuths and the inclinations. Do you assume that the inclinations of both B-measured and B-potential remain nearly constant during the flare? Please either remove references that the field becomes more potential during the flare, since you are not comparing directions of B-measured to B-potential, or add more details on how you compare directions of B-measured to B-potential.

3. The title fonts of several figures (e.g. y-titles of Figure 3) are too small. Increasing them would make the manuscript more readable.

Specific comments:

> Taking advantage of the unprecedented resolution
> provided the 1.6-m Goode Solar Telescope at Big Bear Solar Observatory,
> we discovered a transient rotation of vector magnetic fields,
> about 12° - 20° counterclockwise, associated with a flare.

This phrase might be confusing since rotation of vector magnetic fields has been already reported before. E.g. Petrie (2013) found that in the case of the 15 February 2011 X2.2 flare in NOAA 11158, abrupt untwisting forces occurred in two important sunspots located at opposite ends of the main neutral line or Wang et al. (2014) found that the shear flows at the neutral line and circular motions at the two neighboring sunspots underwent sudden changes during this flare. Please rephrase the abstract to underline the fact that you are talking about transient versus permanent changes.

>and other wild eruptions

Please be more specific. What kind of "wild" eruptions are implied?

>In addition to the brilliant radiation

Please replace "brilliant" with a more specific word or remove it.

>significant changes of the magnetic fields have been observed in the literature.

Please add references to the literature.

> An M6.6 flare

According to GOES it was an M6.5 flare (see e.g.

http://www.lmsal.com/solarsoft/latest_events_archive/events_summary/2015/06/22/gev_20150622_1739/index.html). Please correct.

- > Therefore, by measuring the difference between the azimuth
- > angle of a magnetic field extrapolation based on the vector
- > field and a corresponding potential field extrapolation based
- > on LOS magnetic fields, we can determine whether the twist of the loop is
- > increased or decreased.

Please describe how do you use the LOS magnetic fields to derive the potential field. Typically it is the radial component of the magnetic field that is used to derive B-potential.

Figure 1. Add figure captions to each panel;

>This disturbance of azimuth angle along a narrow ribbon is co-spatial
>and co-temporal with the flare emission seen in H α
See General comment 1. The authors only show a comparison between H-alpha and the azimuth disturbance at 18:00. Hence please either add a side by side comparison of both fronts during the whole flare evolution to confirm the sentence above or add a reference to a specific time at Figure 1, where such a correspondence is demonstrated.

>In the lower right panel of Figure 1, a reversed running difference map of
>azimuth angle is shown.
Please add "at 18:00:12 UT" at the end.

>Figure 1.
Please add captions to each panel.

>On average, the width of the azimuth angle is about 570 km, which is
>comparable to the size of dark ribbon
>in helium 10830 \AA [35].
For clarity please explicitly state the size of the dark ribbon in helium 10830 that you find comparable to the width of the azimuth angle.

>Figure 2:
For clarity I would recommend the authors to add captions to different panels (e.g. some information distinguishing second and third rows, caption to the first row, e.g. "Difference H-alpha image").

>Figure 3:
R1, R2, R3 and Ref captions are hard to see in the top right panel.

>In particular, we see that the azimuth peak coincides with the H α emission
>(dashed curve), based on the results shown in Figures 1 & 3.
See General comment 1. Figure 3 only shows temporal correspondence between H-alpha peak and azimuth peak for R3. Hence please correct the sentence to "We find that the azimuth peak in R3 coincides with the H α emission (dashed curve)" and add the time when the two coincide.

>Only at the flare peak time, the measured azimuth approaches the potential
>level suggesting that the measured field becomes more potential due to the
>sudden rotation.

See general comment 2.

Assumption that the field becomes more potential implies that the angle between the measured field and the potential field decreases. However here you only compare the azimuths of the two, not the azimuth and the inclination. Or do you assume that the inclinations of both B-measured and B-potential remain nearly constant during the flare?

Please either remove "suggesting that the measured field becomes more potential due to the sudden rotation," since you are not comparing the full B-measured to B-potential or add more details on how you compare directions of B-measured to B-potential.

- > In summary, we present a clear transient change of azimuth angle,
- > associated with propagating flare ribbons which are footpoints
- > of 3D magnetic loops.

See General comment 1. The correspondence between flare ribbons and the azimuth disturbances has been demonstrated at only one time. Please either modify the phrase above or add similar comparisons at other times and locations that would demonstrate that ribbons and azimuth disturbances have the same dynamics.

- >2) The strong correlation between the azimuth transient and flare ribbon
- >front indicates that the energetic electron beams are very likely to
- >be the cause.

See previous comment and the General comment 1.

- >3) The change of magnetic field makes them more potential indicating a process of energy
- >release (untwist) of the flare loop.

See General comment 2.

- >However in our case there was no emission.
- Please explicitly specify what is meant here by "emission"

>Movie:

Please, for reference, add the magnetic field contours to the movie, as is done in the top left panel of Figure 3. It would be also helpful if the authors used the same color tables in Figure 3 and the movie.

Reviewer #2 (Remarks to the Author):

The authors have clarified some the most important issues regarding the inversion procedure and they have improved the manuscript. In my opinion, the paper can be published in the present form.

Re: Revisions for manuscript NCOMMS-17-08380A

We would like to thank the anonymous referees again for very helpful comments and suggestions. All of the suggested modifications and reformatting (to comply with format requirements of Nature Communications) have been made accordingly and highlighted in the revised manuscript in PDF format. Below are the detailed responses to the referee reports:

Reviewer #1 (Remarks to the Author):

General comments:

1. The correspondence between flare ribbons and the azimuth disturbances has been demonstrated at only one time (Figure 1). Please either modify the manuscript to reflect the fact that you found correspondence at only one time or add similar to Figure 1 comparisons at other times and locations that would demonstrate that ribbons and azimuth disturbances have the same dynamics. For example, a side-by-side comparison of ribbon vs azimuth space-time plots across the slit perpendicular to the ribbon front or comparison of 1) color map of ribbon propagation with color demonstrating the time of the initial ribbon brightening overlaid on the contours of the magnetic field and 2) color map of azimuth disturbance with color demonstrating the time of the initial azimuth disturbance with contours of the magnetic field, would be helpful.

Answer: Thanks for this important comment. We add to two panels of time-distance diagrams in Figure 1 to compare the positions of H α ribbon front and azimuth angle changes.

2. Assumption that the field becomes more potential implies that the angle between the measured field and the potential field decreases. However in Figure 3 you only compare the azimuths of the two, not the azimuths and the inclinations. Do you assume that the inclinations of both B-measured and B-potential remain nearly constant during the flare?

Please either remove references that the field becomes more potential during the flare, since you are not comparing directions of B-measured to B-potential, or add more details on how you compare directions of B-measured to B-potential.

Answer: We changed the statement to ‘the measured azimuth angle becomes closer to that in the potential field.’ The comparison of B-measured and B-potential is in principle made in a 2D manner, x-y plane, because the measured B_z component remains constant. This point is confirmed by Figure 5, in which the Stokes V component (the major factor in determining B_z) did not vary during the flare. This flare occurred very close to the disk center, so after deprojection, B_z and B_{l₀₅} are identical. The potential extrapolation is based on B_{l₀₅} only and therefore does not vary much during the flare. Among the six components of B_x-measured, B_y-measured, B_z-measured, B_x-potential, B_y-potential and B_z-potential, only B_x-measured and B_y-measured changed. The azimuth angle variation reflects the changes of B_x-measured and B_y-measured. So the 2D comparison is a good

proxy of the 3D configuration. We add this discussion to the 'Results' section.

3. The title fonts of several figures (e.g. y-titles of Figure 3) are too small. Increasing them would make the manuscript more readable.

Answer: The font sizes in Figures 3, 4 & 5 have been increased. The display size of Figure 1 is smaller than the actual size of this figure, because the length of the figure and its caption is over the size of a letter page.

Specific comments:

- > Taking advantage of the unprecedented resolution
- > provided the 1.6-m Goode Solar Telescope at Big Bear Solar Observatory,
- > we discovered a transient rotation of vector magnetic fields,
- > about 12° - 20° counterclockwise, associated with a flare.

This phrase might be confusing since rotation of vector magnetic fields has been already reported before. E.g. Petrie (2013) found that in the case of the 15 February 2011 X2.2 flare in NOAA 11158, abrupt untwisting forces occurred in two important sunspots located at opposite ends of the main neutral line or Wang et al. (2014) found that the shear flows at the neutral line and circular motions at the two neighboring sunspots underwent sudden changes during this flare. Please rephrase the abstract to underline the fact that you are talking about transient versus permanent changes.

Answer: This part of the abstract has been rephrased to emphasize the magnetic change in this study is transient.

>and other wild eruptions

Please be more specific. What kind of "wild" eruptions are implied?

Answer: We changed this part to: and other wild eruptions such as coronal mass ejections (CMEs)

>In addition to the brilliant radiation

Please replace "brilliant" with a more specific word or remove it.

Answer: The word "brilliant" has been removed.

>significant changes of the magnetic fields have been observed in the literature.

Please add references to the literature.

Answer: We added 6 references.

> An M6.6 flare

According to GOES it was an M6.5 flare (see e.g.

http://www.lmsal.com/solarsoft/latest_events_archive/events_summary/2015/06/22/gev_20150622_1739/index.html). Please

correct.

Answer: It is corrected.

> Therefore, by measuring the difference between the azimuth
> angle of a magnetic field extrapolation based on the vector
> field and a corresponding potential field extrapolation based
> on LOS magnetic fields, we can determine whether the twist of
the loop is
> increased or decreased.

Please describe how do you use the LOS magnetic fields to derive the potential field. Typically it is the radial component of the magnetic field that is used to derive B-potential.

Answer: This part is revised to:

“To determine whether the twist of the loop is increased or decreased, the difference between the azimuth angles of extrapolated vector and potential fields are calculated, in which the latter was extrapolated with the Fourier transformation method^{24;25} (Alissandrakis1981,Gary1989). Before the extrapolation, the LOS magnetograms were transformed from the image plane to the heliographic coordinates to remove the projection effects.”

Figure 1. Add figure captions to each panel;

Answer: Captions are added.

>This disturbance of azimuth angle along a narrow ribbon is co-spatial

>and co-temporal with the flare emission seen in H α
See General comment 1. The authors only show a comparison between H-alpha and the azimuth disturbance at 18:00. Hence please either add a side by side comparison of both fronts during the whole flare evolution to confirm the sentence above or add a reference to a specific time at Figure 1, where such a correspondence is demonstrated.

Answer: Time-space diagrams are added to show the temporal correlation between H α ribbon front and azimuth change.

>In the lower right panel of Figure 1, a reversed running difference map of
>azimuth angle is shown.

Please add "at 18:00:12 UT" at the end.

Answer: The observing time "at 18:00:12 UT" is added to the end of this sentence.

>On average, the width of the azimuth angle is about 570 km, which is
>comparable to the size of dark ribbon
>in helium 10830 \AA [35].

For clarity please explicitly state the size of the dark ribbon in helium 10830 that you find comparable to the width of the azimuth angle.

Answer: The size of dark ribbon in helium 10830 is about 340 - 510 km. This sentence is changed to "..., which is comparable to the size of dark ribbon (340 - 510 km) in helium 10830 \AA ".

>Figure 2:

For clarity I would recommend the authors to add captions to different panels (e.g. some information distinguishing second and third rows, caption to the first row, e.g. "Difference H-alpha image").

Answer: The caption is revised to:

“First row: Sparse running-difference maps of azimuth angles, taken at three representative times. Second row: Azimuth angle profiles along the top slit shown in each image on the top panels and the corresponding Gaussian fits. Third row: Azimuth angle profiles along the top slit shown in each image on the top panels and the corresponding Gaussian fits. The FWHM, derived from the fitting, is used as the width of azimuth ribbon, which is about 570 km on average.”

>Figure 3:

R1, R2, R3 and Ref captions are hard to see in the top right panel.

Answer: The font sizes of those labels have been increased.

>In particular, we see that the azimuth peak coincides with the H-alpha emission

>(dashed curve), based on the results shown in Figures 1 & 3.

See General comment 1. Figure 3 only shows temporal correspondence between H-alpha peak and azimuth peak for R3. Hence please correct the sentence to "We find that the azimuth

peak in R3 coincides with the H α emission (dashed curve)" and add the time when the two coincide.

Answer: The light curves of R2 and R3 are added. All the H α light curves are labeled as "H α (R1)", "H α (R2)", and "H α (R3)". The peaks of H α and azimuth angle in each region are correlated. In order to show the H α light curves clearly, we convert them into natural log space and normalized by their own maximum.

>Only at the flare peak time, the measured azimuth approaches the potential level suggesting that the measured field becomes more potential due to the sudden rotation.

See general comment 2.

Assumption that the field becomes more potential implies that the angle between the measured field and the potential field decreases. However here you only compare the azimuths of the two, not the azimuth and the inclination. Or do you assume that the inclinations of both B-measured and B-potential remain nearly constant during the flare?

Please either remove "suggesting that the measured field becomes more potential due to the sudden rotation," since you are not comparing the full B-measured to B-potential or add more details on how you compare directions of B-measured to B-potential.

Answer: The explanation is added in the answer to general comment 2 and in the 'Results' section of the main text.

- > In summary, we present a clear transient change of azimuth angle,
- > associated with propagating flare ribbons which are footpoints
- > of 3D magnetic loops.

See General comment 1. The correspondence between flare ribbons and the azimuth disturbances has been demonstrated at only one time. Please either modify the phrase above or add similar comparisons at other times and locations that would demonstrate that ribbons and azimuth disturbances have the same dynamics.

Answer: Time-space diagrams are added to Figure 1 to show the temporal correlation between H α ribbon front and azimuth change. In Figure 3, two more H α light curves are added for R1 and R2. The H α emission peaks at the same time with azimuth increase in all three representative regions (R1, R2 and R3).

- >2) The strong correlation between the azimuth transient and flare ribbon
- >front indicates that the energetic electron beams are very likely to
- >be the cause.

See previous comment and the General comment 1.

Answer: We added two time-space diagrams in Figure 1 and two light curves in Figure 3.

- >3) The change of magnetic field makes them more potential

indicating a process of energy >release (untwist) of the flare loop.
See General comment 2.

Answer: The explanation is added in the answer to general comment 2 and in the 'Results' section of the main text.

>However in our case there was no emission.
Please explicitly specify what is meant here by "emission"

Answer: This sentence is revised to:

However, in our case, there was no emission detected of the NIR line at 1.56 μm . The NIR line intensity profile remains in absorption during the flare.

>Movie:

Please, for reference, add the magnetic field contours to the movie, as is done in the top left panel of Figure 3. It would be also helpful if the authors used the same color tables in Figure 3 and the movie.

Answer: Magnetic contours and the color table have been added/changed to the same as in the top left panel in Figure 3.

Additional changes to comply with editorial policies:

1. A paragraph is moved from 'Introduction' to 'Methods—Magnetic Inversion'.

- 2. Changed of citation format.**
- 3. Shortened abstract to fit the 150-word limit.**
- 4. Added the last paragraph in 'Introduction' as 'a description of the paper's conclusion'.**
- 5. Added a paragraph of disambiguity in the 'Methods' section.**
- 6. Added a section of Supplementary information**

REVIEWERS' COMMENTS:

Reviewer #2 (Remarks to the Author):

I would like to thank the authors for carefully addressing all of the questions from the previous report. Below please find several very minor corrections, mostly typos that the authors might find helpful. The paper can be published afterwards.

Minor comments:

1.

"It is well accepted that solar flares, and other violent eruptions such as coronal mass ejections (CMEs), result from magnetic reconnection occurring in the corona."

The authors might want to correct the phrase above in the following way

"It is well accepted that many solar flares, and other violent eruptions such as coronal mass ejections (CMEs), result from magnetic reconnection occurring in the corona"

since there are ideal instabilities, such as kink instabilities, that might also lead to flares/CMEs.

2.

"From the time sequence of azimuth maps, we clearly see a ribbon-like structure moving co-spatially and co-temporally with the flare emission in the H α line..."

The authors might want to rethink the location of this part since it is more relevant to results than to the introduction. The link to the Figure here would be also helpful for the reader.

3. "...the vertical component of magnetic field (B_z) and the extrapolated potential field (B_p , derived from B_z), remain "

Please correct "remain" to "remains".

4. "This movies shows the changes"

Please correct "this movies" to "this movie".

5. "On each frame, the white contours outlines the sunspot umbral areas"

Please correct to "In each frame, the white contours outline the sunspot umbral areas"

Re: Revisions for manuscript NCOMMS-17-08380B

We would like to thank the anonymous referees again for the comments and suggested corrections. All of the corrections have been made accordingly and highlighted in the revised manuscript in PDF format. Below are the detailed responses to the referee reports:

Reviewer #2 (Remarks to the Author):

Minor comments:

1. "It is well accepted that solar flares, and other violent eruptions such as coronal mass ejections (CMEs), result from magnetic reconnection occurring in the corona."

The authors might want to correct the phrase above in the following way

"It is well accepted that many solar flares, and other violent eruptions such as coronal mass ejections (CMEs), result from magnetic reconnection occurring in the corona"

since there are ideal instabilities, such as kink instabilities, that might also lead to flares/CMEs.

Answer: This sentence has been corrected.

2. "From the time sequence of azimuth maps, we clearly see a ribbon-like structure moving co-spatially and co-temporally with the flare emission in the Halpha line..."

The authors might want to rethink the location of this part since it is more relevant to results than to the introduction. The link to the Figure here would be also helpful for the reader.

Answer: Yes, this part is relevant to the results. Actually this part was added in last revision in order to comply with the article templates of Nature Communications:

“- Introduction (<1000 words)..... The final paragraph should be a brief summary of the major results and conclusions. The results of the current study should only be discussed in this final paragraph” (stated by the Editor).

In addition, a link to a figure is not allowed according to the check list of the format requirements.

3. "..the vertical component of magnetic field (B_z) and the extrapolated potential field (B_p , derived from B_z), remain "
Please correct "remain" to "remains".

Answer: This word has been corrected.

4. "This movies shows the changes"
Please correct "this movies" to "this movie".

5. "On each frame, the white contours outlines the sunspot umbral areas" Please correct to "In each frame, the white contours outline the sunspot umbral areas"

Answer: Thanks for the suggested corrections. All the corrections have been made accordingly. Points 4 & 5 are in

the description of the supplementary movie. This part has moved to the cover letter as suggested by the editor.

Additional formatting changes suggested by editor are highlighted using light blue color.